# Perceptions of the uses of routine general practice data beyond individual care in England: a qualitative study

David Wyatt,[1,2] Jenny Cook,[1,2] Christopher McKevitt[1,2]

## ABSTRACT

**Objective** To investigate how different lay and professional groups perceive and understand the use of routinely collected general practice patient data for research, public health, service evaluation and commissioning.

**Design, method, participants and setting** We conducted a multimethod, qualitative study. This entailed participant observation of the design and delivery of a series of deliberative engagement events about a local patient database made of routine primary care data. We also completed semistructured interviews with key professionals involved in the database. Qualitative data were thematically analysed. The research took place in an inner city borough in England.

**Results** Of the community groups who participated in the six engagement events (111 individual citizens), five were health focused. It was difficult to recruit other types of organisations. Participants supported the uses of the database, but it was unclear how well they understood its scope and purpose. They had concerns about transparency, security and the potential misuse of data. Overall, they were more focused on the need for immediate investment in primary care capacity than data infrastructures to improve future health. The 10 interviewed professionals identified the purpose of the database in different ways, according to their interests. They emphasised the promise of the database as a resource in health research in its own right and in linking it to other datasets.

**Conclusions** Findings demonstrate positivity to the uses of this local database, but a disconnect between the long-term purposes of the database and participants' short-term priorities for healthcare quality. Varying understandings of the database and the potential for it to be used in multiple different ways in the future cement a need for systematic and routine public engagement to develop and maintain public awareness. Problems recruiting community groups signal a need to consider how we engage wider audiences more effectively.

## Strengths and limitations of this study

► This study explores lay and professional understandings of a local database of routinely collected primary care data being used for purposes other than individual care.
► This multimethod, qualitative approach provides a textured account of local citizens' and professionals' understandings of the benefits, risks and potential of such a database.
► Observation of the design and delivery of a public engagement event provides insights into the engagement process and priorities of lay groups.
► We include interviews with the often overlooked professionals involved in the management and use of routine patient health data for purposes beyond the provision individual care.
► This research is limited to one database in one locality and, therefore, would benefit from future research on similar databases.

Datalink (CPRD) and its predecessor) and the delivery of care (eg, summary care records) at a national level,[2–8] to date, no research focuses on the public's views of using electronic primary care records for commissioning, service evaluation, public health and as a resource for research within a local setting. This paper presents results from a qualitative study of public engagement work designed to inform local people about a database of primary care data from an English inner city borough. The paper investigates how different groups perceive and understand this database and its data practices.

### Background

Published in 2006, the Department of Health's *Best Research for Best Health* balances the right to free healthcare through National Health Service England (NHS) against a responsibility to contribute to the wider welfare of society by participating in health research.[9] It signalled a step change in how we view the role of the citizen not only as a recipient of care but, coupled with later

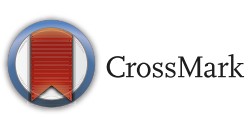

[1]School of Population Health and Environmental Sciences, Faculty of Life Sciences and Medicine, King's College London, London, UK
[2]NIHR Biomedical Research Centre, Guy's and St. Thomas' NHS Foundation Trust and King's College London, London, UK

**Correspondence to**
Dr David Wyatt;
david.wyatt@kcl.ac.uk

## INTRODUCTION

Primary care data are claimed to be an untapped resource in health research.[1] While previous work explores public understandings of and expectations about the use of patient electronic medical records for research (eg, the Clinical Practice Research

changes to the NHS constitution and introduction of the Health and Social Care Act 2012, reimagines the patient and citizen as participants and/or collaborators in the processes of producing health knowledge.[10–12] Contributing to health knowledge in this context does not just mean agreeing to participate in specific trials or projects. One way that this imperative to participate can be achieved is through making routine health data available for research.

The care.data programme, launched in 2013, sought to make routinely collected primary care, hospital and social care records available for audit and research to improve health outcomes.[13] While similar databases have existed for some time, for example, CPRD, care.data generated substantially more media attention and public debate. Many individuals and groups (eg, Healthwatch England) raised concerns about how the use of such data was communicated to the population and the fear of the potential misuse of such data.[14–17] Others, however, saw care.data as essential for the future of health research and argued that fears were the result of scaremongering.[18 19] In July 2016, the programme was cancelled without fanfare or explanation following the publication of the *Review of Health and Care Data Security and Consent.*[20]

The introduction of Fair Processing Notices (FPNs) in 2014[21] and the accompanying Fair Processing Strategy[22] established a formal requirement for health and social care organisations to consult patients on the use of their data for purposes beyond individual care. Existing research has focused on involving citizens in research projects, the impact of public and patient involvement and or engagement, the (lack of) methodological and theoretical underpinnings of such practices and the lack of conceptual clarity around 'engagement', 'involvement' and 'research'.[23–28] Engaging groups about repurposing routine data has received less attention. Existing work that explores public understandings of and expectations about the use of patient electronic medical records for research (eg, CPRD and its predecessor) and the delivery of care (eg, summary care records) at a national level suggests that the public is often positive about data sharing practices for healthcare provision.[3 4] This positivity decreases slightly when data are anonymised and shared for research.[3 4] Citizens, according to this literature, balance the potential harm of participation against the potential benefit of such participation. Although the NHS is often trusted by citizens, this does not necessarily extend to trust in the process of data sharing for research.[2–8]

The focus on national projects and the engagement of the lay public in discussions of data sharing and data usage initiatives ignores the work taking place at a local level. It also overlooks those groups that are part of the process of developing, managing or using the resulting data. This paper, in part, addresses this gap, exploring the process of engaging local people in discussions about the use of routinely collected general practice data for purposes beyond delivering individual care.

## The Database

First developed in 2006, the "Database" (used throughout as a pseudonym for the case studied) collates the data from targeted fields within electronic primary care records (not the free-form text notes) from all general practices in an English inner city borough. An opt-out consent process is used. Clinical and demographic data are accessible to researchers, commissioners and public health professionals to complete analysis that can improve health through research, be that from research itself, public health practice, service evaluations or service commissioning decisions. The patient's unique NHS number is passed through a complex algorithm to generate a pseudonym so that the NHS records are for all intent and purposes anonymous, but records from different parts of the NHS on the same person could be linked together. The Database currently holds the records of approximately 350 000 patients and is managed by the local Clinical Commissioning Group (CCG), the NHS structure that is tasked with planning and commissioning health services in specific geographical locations. The Database is overseen by a Steering Group made up of members from the local government public health department, the CCG, a local university and a regional Healthwatch, the body responsible for the inclusion of the wider public in discussion on health and healthcare decisions at a local (through regional offices) and a national level.[11]

In 2015 and in line with the introduction of FPNs discussed above, the Database leads were required to undertake more substantial public engagement. Previous efforts were largely limited to notices about the Database displayed in general practitioner (GP) surgeries. The Database Steering Group commissioned a university, an NHS trust and a local Healthwatch to design and deliver the engagement strategy. The project aimed to engage people registered with local general practices to inform them about the Database, how routine data can be used for research and quality improvement and to listen to the benefits and concerns individuals identify about the Database.

## METHODS

We adopted a multimethod, qualitative approach to explore the process of engaging local people in discussions about the use of routinely collected general practice data for purposes beyond delivering individual care. All data were collected by DW, an experienced qualitative researcher with no prior connection to the Database.

Figure 1 documents the research process, its separation from the engagement process and a brief breakdown of the different project elements over time. DW observed the work undertaken by three individuals—a university researcher, an NHS engagement officer and a local Healthwatch representative—who were tasked with collaboratively developing and delivering the engagement project commissioned by the Steering Group. He observed all meetings that took place to plan the

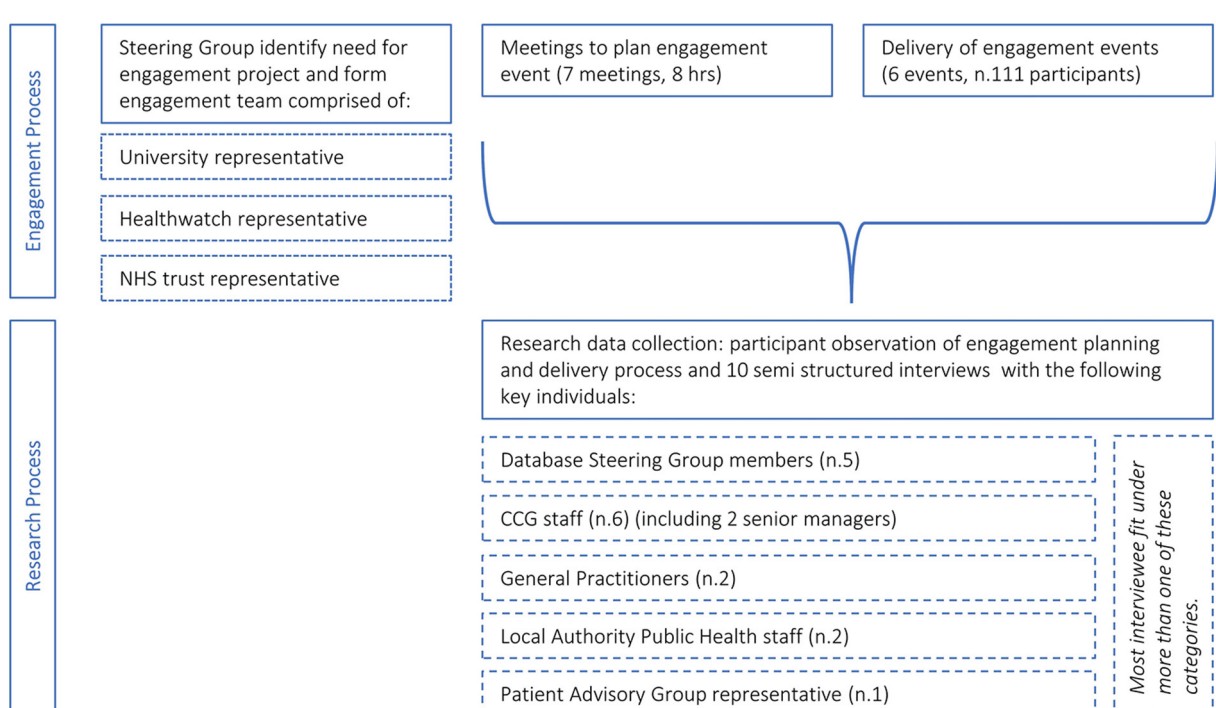

**Figure 1** Breakdown of research and engagement processes.

engagement event, along with the event itself, which was repeated six times to different groups. The event used deliberative engagement methods.[29] Deliberative engagement aims to facilitate a meaningful discussion on topics where the participants may not have a clear, existing understanding of the issue or object being covered. It provides space for participants to 'consider relevant information, discuss the issues and options and develop their thinking together before coming to a view'.[29]

The resulting events aimed to provide a site for members of the public to obtain information and contribute to discussions about how local primary care data are being used beyond individual care.[i] Meeting and event observation data were recorded through contemporaneous note-taking with attention paid to the ways the Database was discussed and the points and questions raised by the public.

In-depth, semistructured interviews with a purposively sampled group of key individuals involved in the Database supplemented the observational data. Interviewees included the Database Steering Group, local CCG sponsors, university research staff, members of the local council's public health department and/or the lead for a local patient participation group (see figure 1). All members of the Steering Group were invited to participate by email. Many invitees fitted into more than one of the groups stated above. Interviews were audio-recorded. Box provides an outline of the topics discussed.

Interviews, transcribed verbatim, and observation notes were uploaded to NVivo10 and analysed by DW using thematic analysis. Data saturation was achieved for interviews. Data were open coded first for descriptive and then analytic accounts.[30] Data and the coding tree were then discussed by all authors, and themes were refined.

## RESULTS

In total, 11 hours of meetings and 6 hours of events (across all six events) were observed. Ten semistructured interviews with professionals were also completed, varying in length from 23 mins to 40 mins (averaging 33 mins). In total, 15 individuals were invited to participate voluntarily by interview. Of those, five declined to participate either by emailing to state that they support the research but are unable to participate in the project's timeframe (n.2) or by ignoring the request and subsequent emails (n.3).

> **Box    Outline of interview topics**
>
> ► Understandings of the Database.
> ► Experiences of using the Database (if relevant).
> ► Experiences of speaking to the public about the Database.
> ► Organisational priorities.
> ► Future of health and healthcare.
> ► Benefits of the Database.
> ► Patient concerns about the Database.
> ► Database governance.

---

[i]In addition to these six events, the engagement team also produced an information leaflet. The leaflet is not discussed in this paper.

| Table 1 | Breakdown of participating groups | | |
| --- | --- | --- | --- |
| **Participating groups** | **No of participants** | **Female** | **Male** |
| Patient participation leads from local primary care practices (pilot) | 9 | 6 | 3 |
| Healthwatch trustees' meeting | 17 | * | * |
| Educated patients' group | 10 | 9 | 1 |
| Young peoples' sexual health service patient advisory group | 4 | 1 | 3 |
| Community mental health group | 23 | 10 | 13 |
| Older people's community group | 48 | 34 | 14 |

*Gender breakdown not obtained at this meeting.

## Identifying 'the public'

The engagement team decided to target community groups in the Database locality (with participants likely to be registered in relevant general practices) to discuss the Database during their regular meetings. Healthwatch provided access to five groups from their existing contacts. The university and NHS trust contacted 13 organisations where no existing relationship was held. Of these, one did not respond and two declined to participate. All others expressed some interest in further information, but only one group, an older people's community group, agreed to participate (see table 1). There were funds available to deliver more of these events had more groups been willing to participate. As such, only one group was not known by an engagement team member prior to the project. The older people's community group was also the only group without specific interests in health. Using existing meetings means that participants had not volunteered to attend a specific meeting about the Database. No groups that are openly sceptical of health data sharing, such as MediConfidential, were invited to participate.

## Participants and events

The participating groups varied in size and gender distribution (see table 1). A total of 111 individuals participated in the events. Each event aimed to provide a site for members of the public to contribute to discussions about how local primary care data are being used beyond individual care. Participants were given a 10-min presentation on what the Database is, the data collected and how it is collected. The presentation also described who sits on the Steering Group, their role in governing the use of Database data, who can access the data, some data management risks, how these risks are averted and the process through which individuals and organisations must go to gain access. This was also mapped out on a whiteboard and remained visible for the duration of the event and, where necessary, provided a reference point for the discussion.

The presentation was followed by two examples of how research produced using Database data has contributed to clinical knowledge. Participants were then given time to discuss the Database and raise questions. They were also asked about their thoughts on this use of patient data, any concerns they have and any research questions they think should be pursued.

## Public understandings of the Database

The participating groups listened carefully to the engagement team's presentation and were generally positive about the Database. For some, the presentation and discussion put them at ease, 'I have no more concerns after the presentation and discussion. I think this is a great and progressive idea' (field notes, event 1). Others affirmed the Database through more general claims of its potential, 'It will benefit us as it will help to improve the healthcare service in general' (field notes, event 5). Despite this overwhelming positivity, some concerns remained, namely: the lack of transparency (they did not know about the Database); Database security and the potential (mis)use of Database data for other purposes by third parties, like the police and insurance companies.

Beyond these concerns, participants discussed wider issues they deemed relevant. They queried: the quality of the data itself and suggested how it could be improved (six events), the source of and level of funding for the Database (four events) and the time burden placed on GPs to facilitate data sharing (five events). Participants' accounts led to wider problems being raised, often relating to primary care capacity. Some questioned whether the funding used to support the Database should be redirected to invest in patient care in general (one event) or in increasing primary care provision in particular (four events). Difficulty accessing GP appointments was raised explicitly (two events) and generated substantially more debate than the Database.

The engagement team specifically acknowledged the potential to link the Database to other datasets through the patient's pseudonymised NHS number. Participants across all events were surprised that generic databases within the NHS are not already linked up. Data linkage for research was not discussed further.

When asked for future research areas or questions, participants often focused on issues that could not be investigated using the Database. In two cases, the data needed to look at specific correlations were not held in the Database. Another relied on individuals being identifiable. The proposed research ideas suggest some confusion around the scope and potential of the Database.

## Understanding data ownership

Participants discussed primary care data as if it were the property of the individual. This idea was perpetuated by the engagement team in their presentation of the Database at the deliberative events, 'In a nutshell, it is a way of

GPs sharing *your* data for the benefit of the community' (field notes, event 4), as it was in the language of those attending the events, 'I do not mind who has access to *my* records, so long as it is not a commercial company' (field notes, event 1). This claimed ownership led, in part, to some disquiet over the use of health data without the patient's knowledge or explicit consent, '[The Database] isn't new so why are you telling us about it now? Does this mean that *our* data has already been used?' (field notes, event 3). Others were cynical, 'it's been going on for so long, it doesn't really matter what I think' (field notes, event 3). Together, these quotes reiterate this lack of transparency about the Database but suggest for some it led to a sense of disenfranchisement. Irrespective of the legality of ownership, participants saw these personal records as their own property.

### Professionals' understandings of the Database

Professionals came from a variety of backgrounds. When asked to describe the Database and its purpose, they tended to emphasise slightly different aspects:

> I mean [the Database] has been around a long time in (the area), led or bred from our public health team[… ] It has always been seen as a useful suite set of information for us to commission appropriately, for public health to advise us in terms of health inequalities appropriately, so it's useful data for us to have. (Interviewee 7, CCG).

> You know, the use of it, the collection of stuff that goes into [the Database] and the benefit that comes out of it, should be something that clearly is visible for access, and should inform delivery of good and appropriate care. But, you know, the other benefits clearly for research and for the monitoring and evaluation and, you know, forward planning the bits. (Interviewee 2, GP and CCG).

The first account places emphasis on the commissioning and public health potential of the Database in data-driven decision making. The second quotation stresses these two elements plus research and the monitoring and evaluation of care. These emphases align to their roles as CCG manager and GP and CCG board member respectively, but the different objectives of research, commissioning, public health and service evaluation were not positioned in opposition to each other.

### 'Data'

Both the community group participants and professionals saw potential in 'data' in general and the Database in particular. It was the professionals, however, who articulated this more expansively in terms of improving public health, providing data for research and commissioning decision making. Professionals often focused on having access to certain types of data, with accounts predicated on an assumption that good data have the potential to positively impact on the health of the local population:

> [G]eneral practice information is the best contemporaneous source of information that we have on the population and the health of the population [… I]t's much better if that can also be segmented, not just by health condition or whatever, but by population group. (Interviewee 10, Public Health).

> I think data is key to everything we do, it underpins it really. And I think whenever you produce a business case there will be other people that look at the finance, there will be people that look at the narrative. But then most people look at the data and what is it telling us and what do we need to be doing to change things. (Interviewee 5, CCG).

Having these data and being able to disaggregate by specific variables renders it 'the best contemporary source of information we have on the population and the health of the population'. Both interviewees 10 and 5 present accounts of how more detailed and relevant data are useful in developing health knowledge, designing interventions and making management and commissioning decisions. Such pushes to collect these data and other relevant datasets present a view of the Database as becoming a more central part in future CCG and research practices. They also reiterate a vision of *data* as something powerful and full of promise. This thinking was often grounded in a potential for linking Database data to other datasets to get a fuller record of individual patients' health:

> [… T]he real potential of primary care data is that it should be linked with [Accident and Emergency department] data and secondary care. And at present, it cannot be without going through the huge rigmarole of Section 251 applications. And they are very labour intensive. And you have to inform the patients that you're doing it and it's a big deal. And it would be very good to find a way to do that that respected Caldicott 2 principles, respected the right of individuals to know what's happening to their data but, at the same time, gave us the potential to look at why do patients attend A&E when their GP surgery is open? (Interviewee 4, Database Steering Group).

> [T]heoretically hospital records could be put through the same sort of pseudonymisation to link them up. But this link is the sort of really contentious part, isn't it? [… T]hree years ago you would have been able to just do that. And we'd have done it and everyone would have been happy and we'd have, you know, a fantastic output from it. And I think we can still get there, but it's just, you know, it's delayed everything by a couple of years definitely. (Interviewee 5, CCG).

These accounts present a future-focused view of the Database. Its current form, as these interviewees infer, is only the start of how data and the Database can improve health and healthcare delivery. Professionals discussed data linkage with overwhelming positivity. However, in the time between Database inception and being prepared to start data linkage processes, the revised

Caldicott principles were released, increasing protection for patient data but making data linkage a more complicated process. In particular, it demands public engagement and includes consent requirements, different to those currently used by the Database.[21] Accounts, such as interviewees 4 and 5, position the potential benefits to the population of data linkage against the need to engage citizens and address their concerns instilled in the revised Caldicott principles.

## DISCUSSION

This paper explored lay and professionals' understandings of a local primary care database made up of routine patient data. Participants in the engagement events were generally positive about sharing primary care data to help improve the health of the local population through research, public health, commissioning and service evaluation. They were, however, concerned by the lack of transparency, the potential to misuse data, patient privacy and commercialisation. These findings reflect those from existing studies.[2–8] Participants displayed a level of cynicism towards the engagement exercise, particularly because, like the care.data programme, many participants were hearing for the first time about a database, established in 2006. The debates prompted by the care.data programme foregrounded the issues of transparency and trust. These concerns serve to question the suitability of the opt-out consent process, which assumes that people know and understand that they are participating in research, that they are aware of how participation takes place and that they can decide to withdraw from participation and how to do so. FPNs, as discussed above, require public knowledge of and the incorporation of engagement into initiatives like the Database. The lack of general awareness of the Database and these local research practices suggest this is currently not being achieved. Sustained public engagement to establish and maintain general awareness of individual databases and such data practices is required.

It is noteworthy that data sharing and the reappropriation of data did not seem to capture the interest of local community groups. Despite attempts to contact new groups, the deliberative events engaged a very specific set—five were well versed in health discussions and already known to the engagement team. These 'usual suspects' reflect the practices of the engagement team and may also be emblematic of the interest of groups in discussing health data and data sharing. This Database was presented in the engagement events as a local resource for local solutions. This may also have positively influenced the decision of groups to participate. The limited levels of recruitment of those not already engaged with Healthwatch suggest we need to consider different ways of engaging individuals and groups in discussions on issues of data and data practices. This is made more salient by the shift towards digitising the NHS and the greater future potential to mine these digital records.[31]

One way of improving awareness and engagement could be achieved through the simultaneous use of multiple different forms of engagement, such as more traditional public events accompanied by social media outreach.[32] We identified a disconnect between the purpose of the Database as a tool for improving future health and the current concerns of participants about the quality and quantity of NHS (primary) care provision. Community group participants' accounts emphasise short-term needs over the potential long-term benefits of the Database. They stress the more personal and practical, such as being able to get a GP appointment when necessary. Each of these individuals, unless they chose to opt out, are participating in the Database; they are, in part, research subjects. Rationales for participating in research are complex.[12 33] Studies suggest that although altruism plays a significant part in why people participate in health research, self-interest can often be a more pivotal factor, for example, helping improve their health or that of a family member.[34 35] Participation in the Database is not for the health of the individual but for population-level health. The results of using the data could very well be of benefit to the population but detrimental to an individual. For example, Database data used for commissioning might result in some services, relevant to some individuals, being decommissioned. With rationales for research participation at the very least including some level of self-interest, we need to acknowledge the potential for a conflict of interest between the individual needs of participants and the use of database data to direct and justify decision making. Such possible outcomes need to be acknowledged by and considered within future engagement activities.

The push for participation in health through infrastructures such as the Database is part of an emergent concept of the patient as active and involved in decision making and research, as 'a care manager [and] a co-producer of health'.[25 36] Carter *et al* suggest the Health and Social Care Act 2012 reflects this 'reimagining and re-responsibilisation of patients as active citizens'.[11 12] The inferred responsibility to play your part for the wider population by taking part in research, supported by a framework of opt-out consent legitimised through the revised Caldicott principles,[21] places a very clear expectation on the individual to be a good citizen and participate. The Database, however, is a complex entity to understand. Discussions with lay participants on potential future research suggested a level of confusion on the potential and scope of the Database as a research infrastructure. Professionals gave varying accounts of the purpose of the Database. Patients as active citizens, in this example, are being asked to consent passively (by not opting out) to participate in an infrastructure that remains somewhat ambiguous. To some extent, this is understandable. The Database is infrastructure for future research, not a discrete research project. Nonetheless, its multiple, potential uses complicate consent.

The potential benefits attributed to the Database and of participation in the Database are often predicated on

a buy-in to a big data future. Data linkage may be more difficult now because of the revised Caldicott principles, but it is an important area of development. Professionals foregrounded the potential for data-driven decision making and data improving future health. Community group participants accepted this potential of the database and contributed to discussions with queries about data quality. We were surprised that community group participants did not raise health data breaches reported in the news. Big data practices, and the ease with which digital data sets can move, be copied and shared, have serious implications for research while also raising ethical questions around the appropriate use of such data. Even when completed with patient benefit in mind, using patient data beyond individual care without full and transparent consent can be problematic, especially when that data move into a commercial setting. In recent months, the UK's Information Commissioner's Office deemed that the Royal Free NHS Foundation Trust had not complied with the Data Protection Act by not doing enough to inform patents that their data would be used in research.[37 38] The Royal Free NHS Foundation Trust allowed details from 1.6 million patient records to be accessed by the Google Deep Minds Project to develop and test a process that would allow doctors to identify patients at greater risk of acute kidney injury. Such an example goes beyond concerns about hacking and the deliberate 'misuse' of routine data in general or its use simply for profit. This particular collaboration can be read as a cautionary tale, highlighting the importance of reflexive and ethical thinking in attempts to deliver 'patient benefit' and the need for systematic and consistent patient awareness and engagement work. It also raises questions about how to manage (if one allows at all) commercial access to such data and how the balance between profit and patient benefit is negotiated. All of these questions require engagement of and input from citizens, especially when the purpose of such research and parameters of what citizens are passively consenting to remain ambiguous. To date, while patient and public involvement agendas often focus on specific research projects and priorities, more work is needed to consider and engage citizens in discussions on the potentially vast uses of patient data beyond individual care.

## Strengths and limitations

To date, research has not focused the real-time engagement of the lay public in such infrastructures and often ignores the professionals involved. Our study provides a textured account of participants' understandings, perceived benefits and concerns. It is, however, based on a limited sample of lay participants, with five of six groups well versed in health issues, and of professionals with direct or indirect association with the Database. The research observed an engagement process that was commissioned by the Steering Group of a specific database to raise awareness of that database, its benefits and document public concerns so they could be addressed.

By emphasising some of the benefits, the engagement process may have positively influenced the responses provided by lay participants. The absence of any groups that were actively against using health data beyond the provision of care and the limited sample may affect the coverage of our analysis. Similarly, demographic details of those who took part in the engagement event were not recorded. Such information could help understand who is being engaged with and any differences in patterns of understandings or concerns. Nonetheless, these findings provide useful insights into how individuals and groups discuss health data sharing. With more research relying on existing datasets, this study signals the need for wide-scale engagement, beyond the confines of a specific database, to garner public understandings, opinions and concerns on the use of health data beyond individual care. However, in order to do this effectively, further work is needed to respond to the question, beyond the scope of this paper: how can we best make different groups aware of these data practices and, where appropriate, engage them in discussions on such practices? Answering this question could also provide a site to explore perceptions of and concerns regarding frameworks and limits of consent, key issues identified in our study.

**Contributors** CMcK conceived the research. DW designed the study with critical feedback from CMcK and JC. DW collected and analysed the data with input from all authors in refining analysis. DW drafted the manuscript and integrated critical feedback from CMcK and JC. All authors read and approved the final version of the manuscript.

**Funding** The research was supported by the National Institute for Health Research (NIHR) Biomedical Research Centre at Guy's and St Thomas' NHS Foundation Trust and King's College London.

**Competing interests** None declared.

**Ethics approval** King's College London Research Ethics Committee (LRS14/150982) granted this research ethical approval.

**Provenance and peer review** Not commissioned; externally peer reviewed.

**Data sharing statement** No additional data are available.

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
