## [Reviewer comments · BMJ Open]

ARTICLE DETAILS

TITLE (PROVISIONAL)	Perceptions of the uses of routine general practice data beyond individual care in England: a qualitative study
AUTHORS	Wyatt, David; Cook, Jenny; McKeivitt, Christopher

VERSION 1 – REVIEW

REVIEWER	Siobhan O'Connor University of Glasgow, United Kingdom
REVIEW RETURNED	24-Sep-2017

GENERAL COMMENTS	This paper describes a qualitative study that engaged local people's views on how routinely collected primary care data is used for research and other purposes beyond care and what professionals opinions are on this subject. I suggest some minor revisions outlined below: • Could you provide a brief table outlining the characteristics of those you interviewed and how many of each e.g. gender, age, professional group or did you collect data on this?• Was the age of the public who participated in the engagement events captured at all? It would be interesting to see if there were differences in the views of older versus younger groups towards the database and personal data use or other characteristics that might affect public opinion on this topic.• Do you know if the risks around data management and governance in primary care were presented at the engagement events to raise awareness of this with the public that took part as it could have influenced their responses?• Did any of the professionals who were interviewed comment on the public's view of data ownership that was brought up during the engagement events? It would be interesting to hear their perspective on this.• It would be nice to see some of the discussion section of the paper, in particular the paragraph around Big Data, mention some of the recent breaches of patient privacy for example the Google DeepMind project. I appreciate this wasn't a primary care dataset but is there potential for something similar to occur with the Database and primary care data in general? Will Big Data mean the NHS and primary care practices have to collaborate with large multinationals who have the computing power and analytics expertise to really deliver on Big Data and what could happen to linked patient data in this possible future scenario if shared with commercial companies? http://www.bbc.com/news/technology-40483202
--

	 • Also Big Data now typically has 5 v's – you could add 'Value' and 'Veracity'. Reference number 36 is quite an old one so have a quick read of more up to date Big Data literature and you'll see these being mentioned. • There seems to be a small spelling mistake on page 4, line 41 – "sometimes" – should be "some time"? • There seems to be a small grammatical error on page 6, line 12 – "delivery", should be delivering? Thank you for providing the COREQ checklist as an appendix as it was helpful to refer to when reading the paper.
--	---

REVIEWER	Harriet Teare Researcher in Healthcare and Policy, University of Oxford, UK
REVIEW RETURNED	16-Oct-2017

GENERAL COMMENTS	This is a timely paper exploring a relevant issue relating to health data use. This paper is methodologically quite complicated, with several different elements. It would be helpful to have a bit more clarity on the separate parts and how they fit together (maybe a diagram would help?) to determine who the different groups are – eg to be more explicit about the engagement team being separate from the researchers observing them, and whether the engagement team is made up of a collaboration between the different commissioned groups, or if this just exploring one engagement exercise of several? In particular the first sentence of the methodology is confusing, and would benefit from being broken down to make it clearer exactly what happened. It is interesting that the discussions with the community groups served two distinct purposes – to inform them about the database and raise awareness, as an engagement exercise, but also to gather their views on the database, as a research exercise. It would be helpful to include more detail about the challenges of incorporating these two aims into one exercise, and whether that will have influenced the research findings, depending on how the engagement portion of the events was pitched – potentially persuasively, to show the database was a valuable asset, rather than neutrally to provide a setting for a more traditional focus group discussion. Given the statement on page 9 (line 11) that there was clearly some confusion around the scope and potential of the database, and that this was an engagement exercise intended to inform people about the database, it would be useful to include further discussion about how to improve information delivery in the future, perhaps with other types of event, or other information formats. Please include greater detail of how the professionals were recruited for interview, including how receptive they were to being included in the study, what the recruitment rate was etc, and whether there were any limitations associated with this part of the study, for example a skewed representation of stakeholders or specific voices not included.
---

	The authors could include a more detailed discussion on the consent issues. Given several participants were not aware of the database, does this undermine the use of an opt-out mechanism (if not coupled with a more effective engagement programme)? Particularly in light of care.data, where one of the criticisms of the project was that patients were required to opt-out within a short timeframe, resulting in GP's opting out on behalf of their entire practice (which raises several of its own issues). Was there any discussion on the consent approach in the groups? If so it would be of interest to include this in the results section. The authors could include reflections on what they think needs to happen next to further our understanding of the wider populations views on these issues, particularly given the difficulties in recruiting groups in the first instance. Do they have any recommendations on the next steps for improving engagement tactics locally and nationally for these initiatives?
--	--

VERSION 1 – AUTHOR RESPONSE

Siobhan O'Connor

Comment: Was the age of the public who participated in the engagement events captured at all? It would be interesting to see if there were differences in the views of older versus younger groups towards the database and personal data use or other characteristics that might affect public opinion on this topic.

Response: This information was not captured. We have added this to the Strengths and Limitations section.

Comment: Do you know if the risks around data management and governance in primary care were presented at the engagement events to raise awareness of this with the public that took part as it could have influenced their responses?

Response: Risks and data governance were covered in the presentation. We have added a sentence to make this clear (Participants and Events section).

Hacking was mentioned at one event. Otherwise, no concerns were raised about data management and governance specifically.

We do not have data on participant concerns prior to the event and the deliberative nature of these events foregrounds the need for giving information to participants prior to discussion. We have, however, acknowledged in the Strengths and Limitations section that the link between the engagement event and a specific database (coupled with the event highlighting the benefits of the Database) may have affected lay responses.

Comment: Did any of the professionals who were interviewed comment on the public's view of data ownership that was brought up during the engagement events? It would be interesting to hear their perspective on this.

Response: The interviews with professionals and engagement events took place concurrently.

Steering Group interview participants received feedback from the engagement team public views, but not until after our data collection had finished.

Interviewees were asked about the concerns they have heard from the public during the course of their everyday work – Public views on data ownership were not mentioned.

Comment: It would be nice to see some of the discussion section of the paper, in particular the paragraph around Big Data, mention some of the recent breaches of patient privacy for example the Google DeepMind project. I appreciate this wasn't a primary care dataset but is there potential for something similar to occur with the Database and primary care data in general? Will Big Data mean the NHS and primary care practices have to collaborate with large multinationals who have the computing power and analytics expertise to really deliver on Big Data and what could happen to linked patient data in this possible future scenario if shared with commercial companies? See <http://www.bbc.co.uk/news/technology-40483202>

Response: We have added this example of a data breach to our Discussion and reflected on its implications.

Comment: Also Big Data now typically has 5 v's – you could add 'Value' and 'Veracity'. Reference number 36 is quite an old one so have a quick read of more up to date Big Data literature and you'll see these being mentioned.

Response: We agree. With other amendments, the mention of the 3/5 v's is not no longer included so we haven't added a reference but have removed Laney (2001).

Comment: There seems to be a small spelling mistake on page 4, line 41 – “sometimes” – should be “some time”?

Response: Amended.

Comment: There seems to be a small grammatical error on page 6, line 12 – “delivery”, should be delivering?

Response: This sentence has been deleted as part of other amendments.

Harriet Teare

Comment: This paper is methodologically quite complicated, with several different elements. It would be helpful to have a bit more clarity on:

the separate parts and how they fit together (maybe a diagram would help?) to determine who the different groups are – eg to be more explicit about the engagement team being separate from the researchers observing them, and

Response: We have amended the text in the methods section and created a diagram to document the process.

Comment: whether the engagement team is made up of a collaboration between the different commissioned groups, or if this just exploring one engagement exercise of several?

Response: We have amended the first paragraph of the methods and results sections to provide further information about who was part of the engagement team. We have also added an Endnote stating that a leaflet was also produced but is not discussed here. The events and the leaflet comprised all engagement activity.

Comment: In particular the first sentence of the methodology is confusing, and would benefit from being broken down to make it clearer exactly what happened.

Response: We have reworded this sentence and added further details to the methods section to aid clarity.

Comment: It is interesting that the discussions with the community groups served two distinct purposes – to inform them about the database and raise awareness, as an engagement exercise, but also to gather their views on the database, as a research exercise. It would be helpful to include more detail about the challenges of incorporating these two aims into one exercise, and whether that will have influenced the research findings, depending on how the engagement portion of the events was pitched – potentially persuasively, to show the database was a valuable asset, rather than neutrally to provide a setting for a more traditional focus group discussion.

Response: We have added further sentences on deliberative engagement to make the purposes and practices of this engagement clear. This is important because almost all community participants were not aware of the Database at the start of the event. While the engagement exercise served to provide information about the Database, explain how it works and collect views from the public, the research element was completely independent on this process. We agree that focus groups could have provided additional insights. However, without participants being already aware of the Database, we are unsure how this would have been viable.

Research findings were presented to the Steering Group.

In the Strengths and Limitations we have added an acknowledgement of the potential for the lay responses to be influenced by the steering group commissioning the engagement work and the structure of the engagement where, prior to discussion, the benefits were presented to participants

Comment: Given the statement on page 9 (line 11) that there was clearly some confusion around the scope and potential of the database, and that this was an engagement exercise intended to inform people about the database, it would be useful to include further discussion about how to improve information delivery in the future, perhaps with other types of event, or other information formats.

Response: Addressed under point 16 below.

Comment: Please include greater detail of how the professionals were recruited for interview, including how receptive they were to being included in the study, what the recruitment rate was etc, and whether there were any limitations associated with this part of the study, for example a skewed representation of stakeholders or specific voices not included.

Response: Further details about the interviews have been added, including the response rate and that participation was voluntary (in Results section). We are not sure how one would measure "receptiveness to participate" so have not addressed this specifically in text.

Details of who participated are provided in Figure 1 and we have added a sentence to the Strengths and Limitations section acknowledging the potential skewing of interview participant (to those embedded and involved in the Database)

Comment: The authors could include a more detailed discussion on the consent issues. Given several participants were not aware of the database, does this undermine the use of an opt-out mechanism (if not coupled with a more effective engagement programme)? Particularly in light of care.data, where one of the criticisms of the project was that patients were required to opt-out within a short timeframe, resulting in GP's opting out on behalf of their entire practice (which raises several of its own issues). Was there any discussion on the consent approach in the groups? If so it would be of interest to include this in the results section.

Response: Consent and care.data are now explicitly mentioned in the Discussion. In the events, although the opt out consent explained and discussed, focus rested on developing an understanding of the database itself and of lay concerns; perception of and opinions on the mechanisms of consent were mostly absent. It was acknowledged by some participants but it was not a substantive theme in our analysis. There are clear implications here, particular in relation to the commercialisation of data. This is addressed briefly in relation to point 7.

Comment: The authors could include reflections on what they think needs to happen next to further our understanding of the wider populations views on these issues, particularly given the difficulties in recruiting groups in the first instance. Do they have any recommendations on the next steps for improving engagement tactics locally and nationally for these initiatives?

Response: We have responded to this in part by adding text to the Discussion to stress the need for awareness raising activity to be included in the routine work of such databases. We have also expressed the need to think carefully about (multiple) engagement methods. In the Strengths and Limitations we have also added sentences highlighting the need for engagement to go beyond specific databases and focus on health data practices more broadly, while also acknowledging the need to complete further research on how best to raise awareness and engage lay publics in discussions of data practices.